# Drug Shortages in Albania: Pharmacists’ Experiences and Perspectives

**DOI:** 10.3390/pharmacy12060166

**Published:** 2024-11-07

**Authors:** Delina Xhafaj, Sonila Vito, Alban Xhafaj

**Affiliations:** 1Department of Pharmacy, Faculty of Medical Science, Albanian University, 1001 Tirana, Albania; s.vito@albanianuniversity.edu.al; 2Department of Data and Digitalization, Intrum S.P.A., 20121 Milan, Italy; alban.xhafaj@intrum.com

**Keywords:** drug shortages, pharmacists, healthcare, supply chain, Albania

## Abstract

Drug shortages are a significant global issue, particularly affecting healthcare systems in resource-limited countries such as Albania. Pharmacists play a critical role in managing these shortages, yet little is known about their experiences and perspectives. This study aims to explore pharmacists’ views on the current drug shortages in Albania, identifying the most affected drug classes, and suggesting potential strategies for mitigating these shortages. A cross-sectional survey was conducted with 93 pharmacists across Albania between December 2023 and May 2024. Data were collected using an online questionnaire that addressed the types of drugs experiencing shortages, the causes of these shortages, and pharmacists’ coping strategies. Cardiovascular and central nervous system medications were reported as the most frequently affected, with supply chain disruptions, regulatory hurdles, and low domestic production cited as key contributors. The findings suggest an urgent need for national policy reforms focusing on improving supply chain resilience and boosting the local pharmaceutical production. The pharmacists proposed mitigation strategies, including stricter regulatory oversight, improved communication channels, and increased local production to reduce dependence on imports. These recommendations underscore the study’s contribution to understanding how tailored, pharmacist-informed strategies could strengthen Albania’s healthcare system.

## 1. Introduction

The global perspective on drug shortages is a critical issue that affects healthcare systems around the world, but especially countries with small pharmaceutical markets that face challenges in terms of access to drugs [1]. Since the World Health Organization (WHO) addressed the lack of medicines in its 2017 report, this has been an issue discussed in scientific literature searches or official reports, with a special interest throughout and after the COVID-19 pandemic, continuing to be reported often in the rest of the world [2,3,4]. According to the “Shortages of Medicines in OECD Countries” report, even in rich economies, access to medicine is increasingly affected by shortages [5]. The PGEU (Pharmaceutical Group of the European Union), representing many community pharmacists in every EU country, surveyed shortages from 2019 to 2022 and reported problems with antibiotic supplies every year since then [6]. Important European organizations such as the European Association of Hospital Pharmacists (EAHP) every year prepares a position paper on this issue, requesting the cooperation of interest groups, the creation of a monitoring platform in 2025, and the harmonization of communication at the European level [7].

Different countries have adopted strategic approaches to mitigate drug shortages, particularly those that impact essential medicines. In the United States, for example, the FDA has implemented early warning systems, and collaborates with manufacturers to anticipate shortages and manage stock levels [8]. European countries, such as Austria, Italy, and Spain use national databases that track medicine availability, making it easier to identify and address shortages in real time [9]. Low-income countries like Ethiopia have focused on optimizing inventory management within their hospital and national pharmaceutical supply chains [10].

Albania is classified as an upper middle-income country according to the World Bank, advancing the European Union integration agenda [11,12]. Albania’s pharmaceutical sector has been shaped by significant policy reforms and dynamic market forces in recent years with the 2014 Law on Drugs and Pharmaceuticals Services and its amendments in 2022 [13]. These changes reflect the country’s efforts to align with the standards of the European Union, and to address various challenges in the availability, quality and affordability of drugs, and regulatory reforms to guarantee the quality, safety, and efficacy of pharmaceutical products [14].

The shortage of medicines in Albania is still complex, with the limited data, and is quite sensitive to external and internal factors. International media echoed the phenomenon, claiming that Albania is suffering from a serious shortage of innovative drugs and therapies, with only 3% of the drugs approved by the European Union in the last three years available in the country, leading to calls from pharmacists to improve the way drugs are brought to the market. However, it is not only drugs for rare diseases or cancer that cannot be bought locally, as this statistic also applies to medications for conditions such as depression, or blood and heart diseases [15]. Among the main pharmaceutical categories affected, according to the media, are some antitumor drugs such as Apixaban and Pembrolizumab [16], but there are no national open data or official reports.

Currently, the National Agency for Medicines and Medical Devices (NAMMD) is responsible for the regulation of the pharmaceutical market in the registration process for new drugs, increasing pharmacovigilance and strengthening quality control measures to guarantee safety and the efficiency of pharmaceuticals in the market [17]. In March 2024, this agency made the first attempt to make the reporting of the issue of drug shortages possible by making a short form for any type of reporting available to the entire population and professionals [18]. There is still a lack of a full national register of data collection of reports as well as a special strategy for monitoring and addressing the issue, making the lack of drugs a significant perception and experience for pharmacy professionals who provide service in the market every day.

Pharmacists are important figures in the Albanian health system, constituting not only the last link in the drug distribution chain, but also being in constant contact with patients. Based on the National Basic Register of members of the Order of Pharmacists of Albania, there are a total of 3108 active pharmacists, of which 1730 are technical managers and 1378 are employed pharmacists [19]. There are approximately 109 pharmacists per 100,000 inhabitants in Albania, a figure comparable to the average of European countries, which have 67 to 112 pharmacists per 100,000 inhabitants [20].

According to previous research studies, there is a lack of a standardized definition for drug shortages, identifying at least 56 definitions worldwide, 26 unique definitions in European Union countries, as well as complete shortages in some high-income countries, compare to low or medium [21] as it is in Albania. Important steps have been taken by international authorities to limit shortages as a phenomenon in Europe, such as the creation of the Medicines Shortages Steering Group (MSSG) of the EMA, which is expected to have a stronger role in providing advice in the field [22]. To limit the phenomenon in the countries of the European Union, the obligation of timely notification of drug shortages has been established by law [23], but the dynamics of this phenomenon in different countries are different, including criteria such as monitoring and a standardizing definition [24] or addressing [25]. In studies conducted previously, a drug shortages impact survey was a frequently employed tool that focused on gathering pharmacists’ opinions on drug shortages and their impact on clinical decision-making [26,27,28,29].

Before the pandemic, according to an economic study of 2016, the pharmaceutical market in Albania was regulated and stable [30]. The COVID-19 pandemic has highlighted its vulnerability, leading to drug shortages and delays in the supply of essential drugs. The import of pharmaceutical products followed a downward trend after its unusual growth during the COVID-19 pandemic [31], and now days imports dominate the drug market in Albania with most of the major international brands represented in the country. Fluctuations in currency exchange rates can lead to significant price variations for imported drugs, affecting their affordability and availability [32,33]. Local production has served in recent years to increase the sustainability of the supply chain and mitigate this issue. There is a movement towards investments in domestic pharmaceutical production and research, counting 103 domestic businesses of this kind in the processing of pharmaceutical products in 2022, compared to only 6 businesses registered in 2021 [34].

The main objective of the study is to investigate the phenomenon of drug shortages in the Albanian pharmaceutical market based on the pharmacists’ own experience, as well as to evaluate the pharmacists’ perception of this phenomenon and the impact it has on the provision of healthcare. The study aims to highlight the lack of specific drugs or pharmaceutical groups according to pharmacists where this phenomenon is evident, as well as the factors that influence this phenomenon and the possible significant correlation between them.

## 2. Materials and Methods

This study employs a cross-sectional design to investigate the experiences and perspectives of Albanian pharmacists regarding drug shortages.

In order to facilitate the study design, components of Intelligence Cycle (IC) (Figure 1) have been used for transforming data into meaningful and actionable data including: planning the study to define the need to understand pharmacist sentiment for drug shortages; data collecting by gathering data from territorial pharmacists through a questionnaire; data processing by cleaning and organizing data through extract, transform, and load (ETL) steps; and analysis of the data performed by Microsoft Excel 365 for statistical purposes, as well for reporting to represent data.

### 2.1. Planning

The planning of the study consisted of exploring the necessity of understanding pharmacists’ perspectives on drug shortages through a questionnaire specifically designed in the Albanian language for pharmacists operating in community pharmacies. It included 22 questions, of which 5 were open-ended and 17 were closed-ended questions. The open-ended questions were designed in order to know the perceived causes and impacts of drug shortages. The closed-ended questions aimed to identify through the given suggestions the most common recommendations for addressing drug shortages. The survey was organized in six sections as follows: 1. Demographic information (5 questions); 2. Experiences with Drug Shortages (6 questions); 3. Opinions on Causes and Consequences (5 questions); 4. Mitigation Strategies (3 questions); 5. Overall Perspectives (2 questions); and 6. Comments and Suggestions (1 question).

### 2.2. Choosing and Validating the Questionnaire

Albania’s unique issues—such as dependence on imports, regulatory barriers, and a lack of a formal reporting system—required a tailored questionnaire to accurately capture local challenges. Unlike standardized international tools, our questionnaire focused on regulatory challenges, supply chain issues, and the roles of national institutions in managing shortages. To ensure validity, the questionnaire underwent a multi-step process. First, healthcare experts reviewed it for content relevance, followed by a pilot test with Albanian pharmacists to refine clarity and suitability. Internal consistency was verified with Cronbach’s alpha.

### 2.3. Data Collecting

Primary data were gathered through a structured Microsoft Form questionnaire distributed online with a link to territorial pharmacists across all regions of Albania. The study was conducted over a six months period, launched in December 2023 and closed in May 2024. Five responders were taken as a focus group to be part of the validation process of the questionnaire before it was shared. The validation process aimed to fulfill issues about “Average time to complete” and “Clarity of the questions”.

### 2.4. Analysis of the Data and Reporting

Responses were anonymous, transferred to an Excel spreadsheet, and subsequently analyzed. Data analysis included qualitative and quantitative analysis from the survey. The qualitative analysis involved content analysis of the responses to open-ended survey questions which were analyzed thematically and categorized to identify key themes and patterns. The quantitative analysis included statistical analysis. Closed-ended survey data were analyzed using descriptive and inferential statistics with Microsoft Office Excel 365.

### 2.5. Ethical Considerations

Ethical considerations were paramount in this study. Informed consent was obtained from all participants before the setting of the survey, ensuring confidentiality and anonymity throughout the research process as in Data Protection Principles set out in the General Data Protection Regulations (GDPR) 2016 Act. Permission by the institution was provided to run such research as well.

## 3. Results

### 3.1. Demographic Information

A total of 93 pharmacists participated in the survey, offering a perspective overview of the pharmacy workforce in Albania. The survey pharmacists’ respondents were mostly female, comprising 73.1% of the sample. This gender distribution underscores the significant presence of women in the Albanian pharmaceutical sector, reflecting broader trends in healthcare professions where female participation is often higher (Table 1).

The age range of respondents varied, with the largest group being those aged between 25 and 35 years, making up 51.6%. This distribution indicates a relatively young workforce of responders, with the majority of pharmacist’s responders being in the early to mid-stages of their careers. A significant part (29.0%) had about 10 years of work experience and extensive pharmaceutical industry knowledge. The survey achieved broad regional coverage, with the majority of respondents working in the major Albanian cities, mostly based in the capital Tirana (49.5%), and followed by Durrës (9.7%), Vlorë (5.4%), and the remaining 24.7% from other cities. The distribution highlights the diverse geographic representation of participants across Albania, providing further insights into regional variations in drug shortages and pharmacy operations. (Table 1). More than half of the responders (51.6%) worked as licensed pharmacists in territorial pharmacies, and 38.7% were technical directors of such pharmacies, indicating a strong representation of front-line healthcare providers, as well as significant involvement in managerial and administrative responsibilities. Smaller segments included managers/supervisors (6.5%), assistant or recently graduated pharmacists (2.2%), and pharmacy business representatives (1.1%). This distribution provides a comprehensive view of the varied responsibilities and positions held by pharmacists in the Albanian healthcare system (Table 1).

Table 2 provides a comprehensive summary, linking demographic characteristics with observations and recommended mitigation strategies obtained from the open and closed-ended questions. Younger pharmacists, often newer to the field, reported higher stress levels and called for increased training support, while those aged 30–50 recognized recurring patterns in shortages and recommended proactive inventory planning. Senior pharmacists, with 15+ years of experience, advocated for more robust regulatory measures, drawing from their extensive professional background. Gender differences also influenced perspectives: female pharmacists, who reported higher stress, emphasized improved communication with regulatory bodies, whereas male pharmacists prioritized logistical solutions like streamlined procurement. Geographic location further impacted findings—urban pharmacists, particularly in Tirana, noted frequent shortages due to high demand and suggested larger stock allocations. Pharmacists in the other regions, impacted by distance from central distribution hubs, advocated for regional centers to reduce dependency on central facilities.

### 3.2. Experiences with Drug Shortages

Pharmacists reported experiencing drug shortages across various therapeutic areas. The most affected areas included cardiovascular, nervous system, and respiratory system drugs, where shortages are frequent and have significant impacts on patient care (Figure 2).

More than 53.8% of the respondents indicated that drug shortages occur frequently, and 29% refer to this phenomenon as always happening, with a general perception (45.2%) that effective communication systems for notifying healthcare professionals of shortages is never available (Table 3). By running the Chi-Square Test of Independence, it is seen that there is no significant association between “experiencing drug shortages” and “the effectiveness of the communication system” (*p*-value < 0.05).

### 3.3. Opinions on Causes and Consequences

The open-ended questions provided to elicit the opinions of pharmacists on the causes of drug shortages identified several key causes of drug shortages related to the regulatory barriers, which were highlighted as a primary cause by the national legal entities, and supply chain issues were reported by 22.6% of respondents as a common cause. Some economic factors (such as being a small pharmaceutical market or low profit percentages) were also indicated by 14.0% of the respondents, so financial constraints and pricing policies exacerbate the problem. The lack of information at different stages of the healthcare system was another cause expressed by just 2.2% of responders, while many pharmacists cannot think of possible causes (25.8%) (Table 4).

The opinions on consequences of these shortages are deep, with many pharmacists noting their professional opinions in the open-ended questions on the increase in patient wait times and a need for alternative therapies that may not be as effective or affordable.

According to several pharmacists who answered, there is occasionally a higher chance of drug errors due to shortages (46.2%). According to 62.4% of pharmacists surveyed, shortages always have an impact on workload and stress levels, and they also have an impact on patient costs of healthcare (58.1%). Over 50% of the participants agree that shortages of medications always affect patients’ interactions with pharmacists and have an effect on patients’ health outcomes (Figure 3).

A substantial correlation (*p* < 0.05) has been found between the perceptions of drug shortages and their “impacts on medication errors”, “workload and stress”, “healthcare costs”, “relationships with patients”, and “patient health outcomes.” This illustrates the complexity and wide-ranging consequences of this issue by indicating a connection between the experiences and opinions of those pharmacists regarding medicine shortages.

### 3.4. Mitigation Strategies

Respondents suggested several strategies to mitigate drug shortages through regulatory bodies. The most common responses involved implementing stricter monitoring of drug production (38.0%) and facilitating communication between manufacturers and pharmacies (34.7%), but also increasing transparency through the implementation of fines for undertaking theft or price gouging (18.2%) (Table 5). Respondents were opinioned for the regular publication of drug inventory levels and shortage forecasts to enhance planning and response efforts. When asked about specific improvements in the supply chain, the responses are divided between creating forecasting strategies (38.1%), better management systems of inventory (30.2%), and the implementation of emergency delivery protocols (31.7%) (Table 5). By performing Chi-squared statistic test, it is seen that the distribution of responses is not uniform (*p* < 0.05), indicating that there is not a significant preference for any specific recommendation among the pharmacists surveyed, and that the distribution of responses is not uniform.

When asked about recommendations for policy makers to address the issue of drug shortages in Albania, the suggested technique that received the highest support from respondents (42.6%) was to increase supply chain transparency. Furthermore, 37.5% of respondents suggested that legislators ease import restrictions on necessary medications. Investment in local pharmaceutical production was recommended by 18.4% of respondents. The results of the Chi-test conducted on this data suggest that there is no significant preference or trend in the responses (*p* < 0.05) (Figure 4).

To improve preparation and reaction, respondents demanded that drug inventory levels and shortage projections should be regularly published. Enhancing import procedures was another way to highlight the need for better infrastructure to enable quicker and more dependable importation, and 38% of respondents suggested streamlining the import procedure for necessary medications, which includes assisting and enhancing cooperation between regulatory bodies. Reinforcing local production was the least chosen choice as a recommendation, but merely 18% advocated for investment in local pharmaceutical manufacturing, emphasizing that this is not a favorite strategy (Figure 4).

## 4. Discussion

This study reveals that the general feeling among pharmacists is one concern regarding the current state of drug shortages in Albania. More than half of the pharmacists find these shortages frequently in their everyday work and this aligns with global trends reported by the World Health Organization (WHO) and other international bodies.

Pharmacists have identified drug shortages as a persistent challenge affecting healthcare systems worldwide and the most reported medicaments were those in the cardiovascular therapeutic class followed by nervous system, respiratory system and anti-infective drugs. These therapeutic classes are reported in the Medicine Shortages Survey conducted by the Pharmaceutical Group of the European Union (PGEU) [35], as well as the report of the Organization for Economic Co-operation and Development (OECD) [36] both emphasizing the shortage of antibiotics drugs. This frequency of shortages seems to always have consequences raising the work stress level of the pharmacist and affecting the cost of healthcare for the patient.

Similar concerns are reported even at other studies conducted at community pharmacist in larger and more developed countries [37,38]. The study’s cross-sectional survey used in this study aligns with the methodologies employed by McBride et al., indicating that, similar to Albania, healthcare providers in other countries face frequent shortages in key therapeutic areas, including oncology and cardiovascular drugs [27]. However, unlike in more developed healthcare systems, Albanian pharmacists reported a higher frequency of shortages, particularly due to reliance on imports and a lack of local production capacity.

From the pharmacists’ point of view, the main cause of the drug shortage lies in the regulatory barriers and supply chain issues, and the communication gap between themselves and institutions or suppliers is notable. Not having the proper reporting strategy by the respective National Agency obscured the shortage situation even more. As previously studied by Vogler & Fischer (2020), the policies in different European countries, including Albania, have not yet established a register, and this hesitancy may result from a lack of capacity, as the implementation of such measures requires resources. Aligning with the best practices of reporting, addressing, and mitigating of the European Union’s countries and their legislations could start to improve the current situation. Good examples include France, where the World Health Organization (WHO) is implementing a 3-year project to assist in addressing the challenge of shortages and lack of availability of off-patent antibiotics in human and animal care [39].

Many pharmacists expressed concerns with the existing system, highlighting the need for comprehensive and inclusive policy reforms and better management practices. Key comments included policy and regulation, where respondents stressed the need for more proactive and flexible regulatory frameworks that can adapt quickly to changes in the pharmaceutical market. There was a call for policy makers to engage more closely with healthcare professionals and industry stakeholders to develop effective solutions as well as more informative communication between the regulatory bodies and pharmacists. Other key comments search for sustainability and long-term solutions, including investment in local production and improved supply chain infrastructure, even though the local production is seen to be increasing its numbers in recent years and could have role in mitigating the issue. To achieve sustainability in this process, pharmacists express their opinions on the importance of long-term planning, and constant monitoring in the pharmaceutical supply chain was highlighted as critical to ensuring continuous access to medicines, especially the medicine from the essential list.

Pharmacists in this study proposed several strategies to address drug shortages, emphasizing the need for enhanced regulatory oversight, transparent communication, and investments in local pharmaceutical production. For instance, stricter monitoring and reporting of shortages, similar to systems used in larger healthcare markets, were seen as crucial for timely responses and resource allocation. Improved communication channels between pharmacists and suppliers were also recommended, which could foster greater transparency in stock levels and anticipated shortages.

These mitigation strategies, drawn from frontline experiences, highlight the practical value of this research in shaping national policies. By incorporating pharmacist-driven solutions, Albania can adopt context-specific approaches that align with both local needs and global best practices, ultimately contributing to a more resilient healthcare infrastructure.

The proposed strategies from Albanian pharmacists can be compared to established international practices to underscore potential areas for improvement. Pharmacists suggested stricter regulatory oversight and enhanced communication, aligning with methods used in Europe [9], where real-time shortage databases have proven effective in reducing supply disruptions. The FDA’s early warning system for anticipating shortages [40] also resonates with pharmacists’ calls for proactive measures in Albania, which currently lacks a formal reporting mechanism.

Despite the complexity of the problem at different levels, this first study in Albania gained its strengths by beginning an initiative to identify the experience and prospective of the common issue of drug shortages among pharmacists in their everyday work by developing tools for field data collection, which will significantly help future researchers and policymakers.

Even though the number of the pharmacists that took part in the study was a small part of the total number of pharmacists in Albania (approximately 3%), they were distributed all across the country and were committed to responding to a long questionnaire. Although this study’s cross-sectional methodology allowed for describing relationships between numerous indicators and drug shortages, causal implications cannot be reached because it relies on self-reported data by the responders, which may be subject to biases and personal and professional opinions. A limited access to national open and updated data such as imports and exports of medicine restricted this study from further evaluations. Considering the context of the dynamics and reality of the pharmaceutical market in Albania, different instruments may be used to measure drug shortages in the near future, as well as to adapt strategies from the OECD or EU member countries [5,41].

## 5. Conclusions

This study highlighted the experiences and the perspective of pharmacists in Albanian pharmacies regarding drug shortages, which are stated to be a frequent issue. Pharmacists seem to be concerned about the periodicity of this phenomenon because it affects their stress levels at work and also increases the general cost of healthcare for the patient. There is a general agreement that the main drugs most reported in the shortages are of the cardiovascular, respiratory and nervous system groups.

The primary causes of these shortages identified by pharmacists are related to regulatory barriers and supply chain issues, with a constant lack of formal information by the respective institutions. The lack of a specific section on the National Agency website to publish periodic shortages and formal guidelines on how to deal with specific cases is an issue in the Albanian system which needs to be explored.

The pharmacists advocate for harmonized and on-time communication, including collection, reporting, and feedback of comprehensive data on medicine shortages. There is a need for a national strategy based on the systems of countries that have successfully managed similar issues, involving a deep collaboration with stakeholders, especially pharmacists, to ensure the sustainable availability of medicine in Albania. Implementing such initiatives can offer long-time benefits, such as improved patient outcomes, reduced healthcare costs, and increased trust in the healthcare system.

## Figures and Tables

**Figure 1 pharmacy-12-00166-f001:**
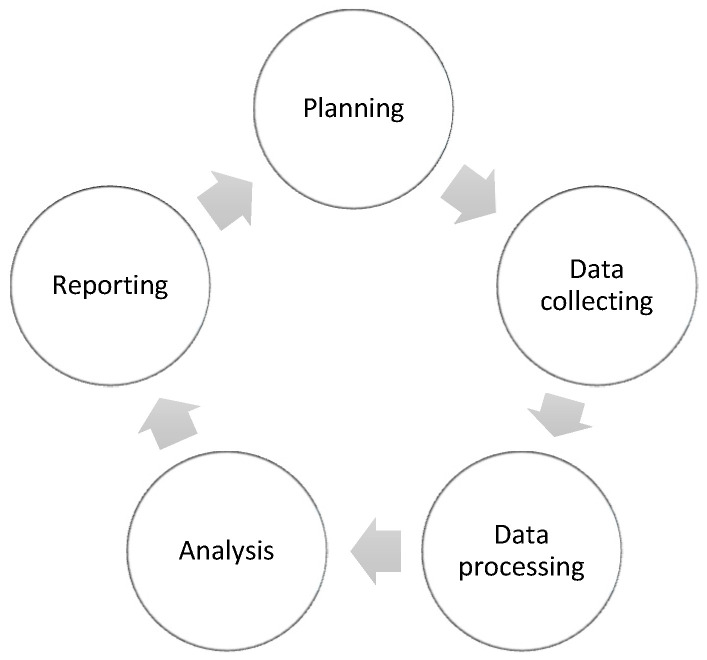
Elements of Intelligence Cycle steps (IC) used in the study.

**Figure 2 pharmacy-12-00166-f002:**
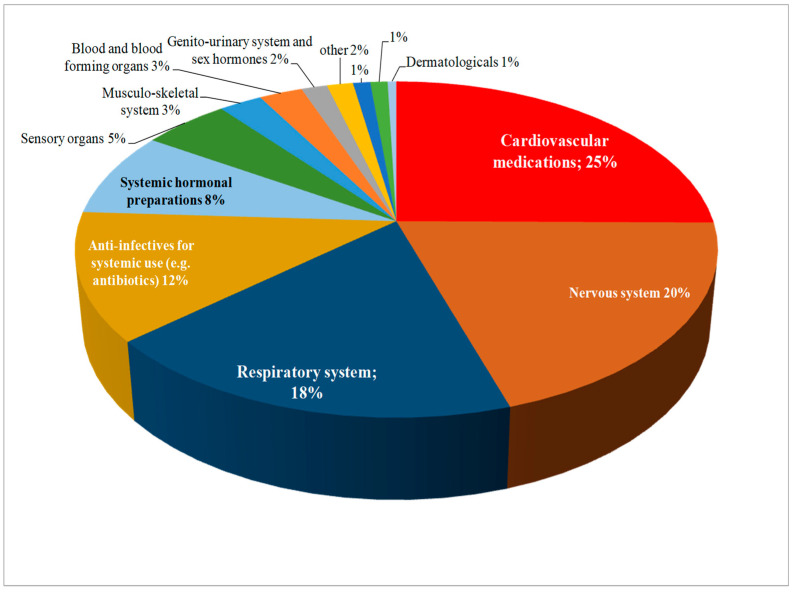
Therapeutic classes of the principal active ingredients included in the shortages reported by the pharmacists in the survey.

**Figure 3 pharmacy-12-00166-f003:**
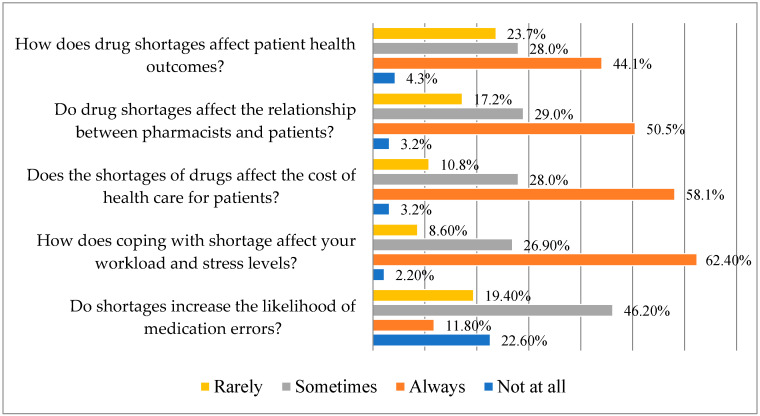
Responses on consequences of the drug shortages (N = 93) (Statistic Value: Chi-squared (χ^2^) 181.34; Degrees of Freedom (df) 12; *p*-value 0.00).

**Figure 4 pharmacy-12-00166-f004:**
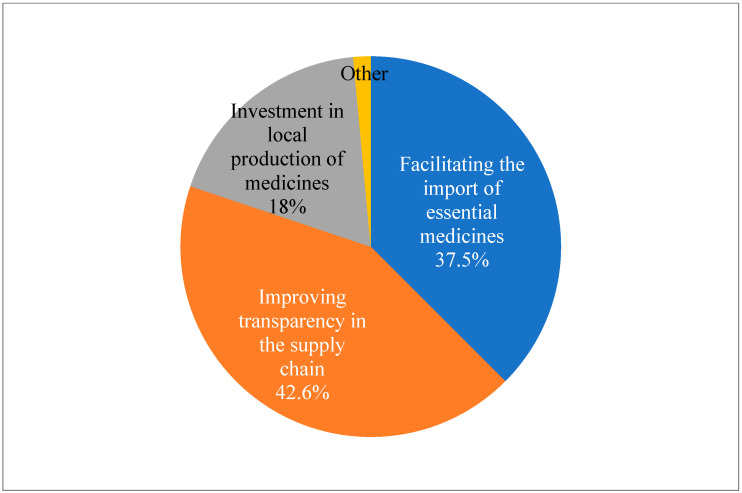
Pharmacist’s recommendations for policy makers to address the issue of drug shortages in Albania (N = 136) (Chi-squared test was conducted to determine if there is a significant difference in the distribution of responses: X2 = 57.9; DF = 3; *p* < 0.05).

**Table 1 pharmacy-12-00166-t001:** Demographic information of the responders.

Total (N)	93	
	Frequency(%)	No.
Gender		
Female	73.1%	68
Male	26.9%	25
Age (years old)		
(1) <25	7.5%	7
(2) 25–34	51.6%	48
(3) 35–44	29.0%	27
(4) 45–54	9.7%	9
(5) >55	1.1%	1
Professional Experience (years)		
(1) 0–1	14.0%	13
(2) 2–5	31.2%	29
(3) 6–10	16.1%	15
(4) >10	38.7%	36
City		
Tiranë	49.5%	46
Durrës	9.7%	9
Vlore	5.4%	5
Elbasan	10.8%	10
other	24.7%	23
Role in the pharmacy		
Assistant pharmacist	2.2%	2
Pharmacist technical manager	38.7%	36
Registered Pharmacist	51.6%	48
Manager/supervisor	6.5%	6
Pharmacy business representative	1.1%	1

**Table 2 pharmacy-12-00166-t002:** Results by age, experience level, gender, and geographic location.

Demographic Feature	Category	Observations on Drug Shortages	Mitigation Strategies Proposed
Age and Experience	Under 30 years (<5 years)	High stress levels; challenges in handling shortages due to inexperience	Enhance training and support for new pharmacists
	30–50 years (5–15 years)	Noticed shortage patterns, especially in cardiovascular drugs	Implement proactive inventory planning based on observed trends
	Over 50 years (15+ years)	Frustration with systemic support; proposed practical, long-term solutions	Advocate for stronger regulatory measures and improved communication systems
Gender	Female	Reported higher stress in shortage management	Focus on improved communication with regulatory bodies
	Male	Concentrated on logistical challenges, such as supply chain management	Suggest streamlined procurement and inventory processes
Geographic Location	Capital	Frequent shortages in high-demand drugs due to larger patient volumes	Increase stock levels and ensure quick distribution to urban areas
	Other Regions	Difficulties accessing essential drugs, compounded by logistical challenges	Strengthen local distribution logistics and ensure rural inventory support

**Table 3 pharmacy-12-00166-t003:** Frequency of the experience of shortage by the pharmacist.

Have You Experienced Drug Shortages in Your Pharmacy?			Is There an effective Communication System for Notifying Healthcare Professionals of Shortages?	
	No. of Responses	% of Responses	No. of Responses	% of Responses
Ever	2.2%	2	45.2%	42
Rarely	1.1%	1	20.4%	19
Always	29.0%	27	4.3%	4
Sometimes	14.0%	13	20.4%	19
Frequently	53.8%	50	7.5%	7
Total	93			

Chi-Square Test of Independence was conducted to show if there is significant association between “experiencing drug shortages” and “the effectiveness of the communication system” (*p*-value < 0.05, X^2^ =103, df = 4).

**Table 4 pharmacy-12-00166-t004:** Reported opinions on causes (clustered by the open questions of the survey).

Opinions on Causes		
	No. of Responses	% of Responses
Supply Chain Issues	22	22.6%
Regulatory Barriers	32	34.4%
Economic Factors	13	14.0%
Information	2	2.2%
Don’t know	24	25.8%
Total	93	

**Table 5 pharmacy-12-00166-t005:** Responses about mitigation drug shortage strategies and improvements.

In Your Opinion, What Role Should Regulatory Bodies Play Primarily in Mitigating Drug Shortages? (Multichoice)			What Specific Improvements in the Supply Chain, If Any, Do You Believe Would Be Most Helpful in Mitigating Drug Shortages? (Multichoice)		
	No.of Responses	%of Responses		No.of Responses	%of Responses
Implementation of stricter monitoring of drug production	46	38.0%	Advanced forecasting methods to predict shortages	48	38.1%
Facilitating communication between manufacturers and pharmacies	42	34.7%	Improved inventory management systems	38	30.2%
Implementation of fines for undertaking theft or price gouging	22	18.2%	Implementation of emergency delivery protocols	40	31.7%
Other opinions	11	9.1%	Other opinions	0	0%
No. Responses	121		126		

Chi-squared test of independence was conducted to show if the observed frequencies for the different recommendations significantly differ from expected frequencies, assuming a uniform distribution (i.e., if each category is equally likely).

## Data Availability

The generated datasets are available from the corresponding author analyzed during the current study but are not publicly available due to the preparation of other manuscripts from the data. Still, they are available from the corresponding author at reasonable request.

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
