# Peer review of "Drug Shortages in Albania: Pharmacists’ Experiences and Perspectives"

_pharmacy, 2024, doi:10.3390/pharmacy12060166_

Round 1
Reviewer 1 Report
Comments and Suggestions for Authors
The authors address an important problem: shortages of pharmaceuticals. Pharmacists in Albania were asked by a questionnaire what and how they perceive, experience and manage this problem. They were also asked about what is needed for the future, like policy recommendations.
The article gives a good overview of the opinions of pharmacists on drug shortages problems in Albania. I have no criticism on the results. As such, the article could be published in “Pharmacy” if the journal is okay with publishing opinions of pharmacists on the drug shortages problem. But the question is how interesting this is? I write this with the background that Pharmacy has published several times questionnaire-based studies without using previous used and/or validated questionnaires.
I advise the authors to consider the following:
* Give overview of similar studies abroad.
* What questionnaires did these similar studies use?
* Why do the authors chose another?
* What is the validity of the questionnaire used? How is that assessed?
* International comparison with studies abroad which used the same study method.
Comments on the Quality of English Language
Minor editing needed.
Author Response
Thank you for your advice on optimizing this research, which is the first of its kind in Albania and needs to be explored further. The authors took your recommendations seriously and implemented them in the manner listed below.
Comment 1: Give overview of similar studies abroad.
Response1: Revised and reflected in the Introduction section, lines 44-51
Comment 2: What questionnaires did these similar studies use?
Response 2: Reflected in the Introduction section, lines 97-99
Comment 3: Why do the authors chose another?
Response 3: Described in the Method Section line 146-154
Comment 4: What is the validity of the questionnaire used? How is that assessed?
Response 4: Described in the Method Section line 146-154, 162-169
Comment 5: International comparison with studies abroad which used the same study method.
Response 5: Revised and described in the Discussion section in lines 357-363
Comments 6: on the Quality of English Language: Minor editing needed
Response 6: The language was revised and edited by the authors
Reviewer 2 Report
Comments and Suggestions for Authors
Thank you for your research: My comments for improvement are:
1. Responders have proposed mitigation strategies. This point could be accentuated in the "Abstract" and should be underlined in the "Discussion" section. 2. The mitigation strategies should be connected to the research-added value. It is not evident if the study only highlights the drug shortage or proposes a remedial action. 3. The way other countries have tackled drug shortage should be described in the "Introduction", and should be compared to the findings in the "Discussion".4. The methodology is scientifically sound. However, the Intelligence Cycle Figure stages should be further analyzed according to the implementation of the method. 5. The results are well-described. However, the results could be presented in light of the demographic features. Comments on the Quality of English Language
Moderate English proofreading is needed.
Author Response
Thank you for your advice on optimizing this research, which is the first of its kind in Albania and needs to be explored further. The authors took your recommendations seriously and implemented them in the manner listed below.
Comment 1Moderate editing of English language required.
Response 1: The language was revised and edited by the authors
Comment 2: Responders have proposed mitigation strategies. This point could be accentuated in the "Abstract" and should be underlined in the "Discussion" section.
Response 2: Revised and reflected in the Abstract section in lines 21-25, as well as underlined in the "Discussion" section 397-408. These reflections were done to give a better idea of the mitigation strategies proposed by the pharmacists who participated in the survey.
Comment 3: The mitigation strategies should be connected to the research-added value. It is not evident if the study only highlights the drug shortage or proposes a remedial action.
Response 3: This part was taken seriously in consideration by discussing it further in the "Discussion" section 397-408, and as explained in the text, this study highlights the drug shortage from the pharmacists perspective and proposes some remedial actions ideas just by this group of professionals and their experience in the working life.
Comment 4: The way other countries have tackled drug shortage should be described in the "Introduction", and should be compared to the findings in the "Discussion".
Response 4: Revised and reflected in the Introduction section lines 44-51 as well as compared the findings in the "Discussion" section in lines 402-408
Comment 5: The methodology is scientifically sound. However, the Intelligence Cycle Figure stages should be further analyzed according to the implementation of the method.
Response 5: The Intelligence Cycle in the Methodology section was further explained and the text was re-organized in order to give a better idea of the steps of IC.
Comment 6: The results are well-described. However, the results could be presented in light of the demographic features.
Response 6: Thank you for the appreciation. A special part of the Results section was dedicated to this argument in lines 202-215 and also Table 2 was used as a tool to describe the demographic characteristics of the Observations on Drug Shortages and Mitigation Strategies Proposed based on the responses.
Comment 7: Comments on the Quality of English Language: Moderate English proofreading is needed.
Response 7: The language used was revised and edited by the authors
Reviewer 3 Report
Comments and Suggestions for Authors
Good effort.
1. Background should be more comprehensive with the significance.
2. Method should be detailed. The description is not enough to understand the concept.
3. Result section: line no 231, 232 .Is it typo?
Graphical 1 and 2 should be more clear and the legend, label etc should be adjusted..
4. Discussion: It should have more references to clarify the issue.
5. The conclusion should be more specific and shortened.
Author Response
Thank you for your advice on optimizing this research, which is the first of its kind in Albania and needs to be explored further. The authors took your recommendations seriously and implemented them in the manner listed below.
Comment 1: Background should be more comprehensive with the significance.
Response 1: The Introduction section was revised and lines 44-51 were added to give a more overall idea of similar studies abroad.
Comment 2: Method should be detailed. The description is not enough to understand the concept.
Response 2: The Intelligence Cycle in the Methodology section was further explained and the text was re-organized to give a better idea of the steps of IC
Comment 3: Result section: line no 231, 232 .Is it typo? Graphical 1 and 2 should be more clear and the legend, label etc should be adjusted.
Response 3: Apologies for the earlier confusion. The authors have made adjustments to address this issue, ensuring that all tables and legends are now aligned with the text.
Comment 4: Discussion: It should have more references to clarify the issue.
Response 4: Proper adjustments were taken, for example adding 37,38,39,41 to give a better overview of the Discussion section
Comment 5: The conclusion should be more specific and shortened.
Response 5: The authors have carefully revised certain phrases in the Conclusion section and believe that maintaining its current length is essential, as it provides a comprehensive understanding of the topic. Thank you for your understanding!
Round 2
Reviewer 1 Report
Comments and Suggestions for Authors
I thank the authors for following up my comments.
Comments on the Quality of English LanguageModerate editing is required.